# Predictive Utility of Composite Child Feeding Indices (CCFIs) for Child Nutritional Status: Comparative Analyses for the Most Suitable Formula for Constructing an Optimum CCFI

**DOI:** 10.3390/ijerph19116621

**Published:** 2022-05-29

**Authors:** Stephen Kofi Anin, Richard Stephen Ansong, Florian Fischer, Alexander Kraemer

**Affiliations:** 1School of Public Health, Bielefeld University, Universitätsstraße 25, 33615 Bielefeld, Germany; stephen.anin@uni-bielefeld.de (S.K.A.); alexander.kraemer@uni-bielefeld.de (A.K.); 2Department of Industrial and Health Sciences, Faculty of Applied Sciences, Takoradi Technical University, Takoradi P.O. Box 256, Ghana; 3Department of Nutrition and Food Science, University of Ghana, Accra P.O. Box LG 134, Ghana; riansong23@gmail.com; 4Institute of Public Health, Charité—Universitätsmedizin Berlin, Charitéplatz 1, 10117 Berlin, Germany

**Keywords:** feeding practice, infant, feeding index, summary index, nutritional status, undernutrition, wasting, pediatric populations, Ghana

## Abstract

Composite child feeding indices (CCFIs) developed from various relevant measures of dietary intake by infants and young children have several potential applications in nutritional epidemiological studies for the development and deployment of precise public health nutrition interventions against child undernutrition. The predictive utility of some CCFIs (computed from varying formulation components) for child nutritional status (stunting, wasting, and underweight) were compared. The purpose of the study was to identify the most suitable among them for possible standardization, validation, and adoption by nutritional health researchers. Using cluster sampling, data from 581 mother–child pairs were collected. Multivariable regression analyses were applied to the data obtained through a community-based analytical cross-sectional survey design. Three of the CCFIs were found to be significantly associated with only wasting (WHZ) from the linear regression models after adjusting for potential confounders and/or correlates. None of the CCFIs (whether in the continuous nor categorical form) was consistently predictive of all three measures of child nutritional status, after controlling for potential confounders and/or correlates, irrespective of the choice of regression method. CCFI 5 was constructed using a dimension reduction technique—namely principal component analysis (PCA)—as the most optimal summary index in terms of predictiveness for child wasting status, validity, and reliability (Cronbach’s α = 0.80) that captured relevant dimensions of optimal child food intake. The dimension reduction approach that was used in constructing CCFI 5 is recommended for standardization, validation, and possible adoption for wider applicability across heterogeneous population settings as an optimum CCFI usable for nutritional epidemiological studies among children under five years.

## 1. Introduction

Child food intake and/or nutrition-related factors are estimated to be associated with about 45% of global mortality amongst pediatric populations under five years [1]. The seemingly irreversible repercussions of poor nutrition on the physical, mental, and psychosocial developments of infants and young children during the 1000-day period (gestation until 23 months) or pediatric populations under two years until substantial height catch-up after five years (during adolescence) is thus dubbed the window of opportunity [2,3,4,5]. The eight core and seven optional indicators of infant and young child feeding (IYCF) practices were conceptualized following consensus in 2007 led by the World Health Organization (WHO) and other global development partners including UNICEF. These indicators were to serve as simple, valid, reliable, and pragmatic pediatric population-level indices of child feeding practices especially in developing countries [6,7,8]. These indicators were mainly aimed at: (i) assessment (to make national and sub-national comparisons and to describe and project compliance trends over time); (ii) targeting (to identify pediatric populations at risk of malnutrition, target nutrition-specific and/or nutrition-sensitive interventions, and make public health nutrition policy decisions about resource allocation priorities); and (iii) monitoring and evaluation (to examine progress in achieving nutrition goals and to evaluate the impact of public health nutrition interventions). Since then, a couple of revisions have been made to these validated indicators [9,10]. The formulas for their measurement and computations have also been standardized to ensure valid comparisons across various heterogeneous study settings, especially in low- and middle-income countries (LMICs) [7].

IYCF indicators are widely utilized within the nutritional epidemiological research (NER) community. They constitute some of the key indicators reported by nutritional epidemiologists, pediatricians, public health nutritionists, and researchers in global, national, regional, and community nutrition study reports. However, NER literature is replete, with study findings of inconsistencies in the predictive utility of the individual IYCF indicators for child nutritional status, contrary to the association theoretically postulated and espoused by the developers and users alike within the NER community [11,12,13]. Furthermore, since the seminal publication of the predictive utility of a composite child feeding index (CCFI) constructed from a combination of individual child feeding indicators (breastfeeding and complementary feeding) by Ruel and Menon [14], a wide array of formulas have been utilized by different researchers to examine the associations between these varied CCFIs and child nutritional status indicators as summarized in Table 1. The NER literature is replete with study findings showing inconsistencies in the predictive utility of the various CCFIs for child nutritional status [15,16]. There is also the lack of a standardized and validated formula for computing an acceptable summary index of infant and young child feeding (IYCF) practices (CCFIs) that is reflective enough of the multidimensionality of optimal child feeding practices for generic application across different study settings, as evinced in the arbitrary and varied formulations from various infant and child feeding indices (ICFI) (Table 1) [11,17]. Besides, there may be the need to rather adopt an integrated approach to addressing the methodological weaknesses inherent in the quest to accurately measure food and/or nutrient intake as an exposure variable of interest to nutritional epidemiologists [18,19]. By harnessing and synergizing the complementary strengths of both the reductionist (Western) and holistic (Eastern or Oriental) epistemological (philosophical) paradigms of food and/or nutrient intake measurement approaches, the association between food intake as the exposure variable of interest and diet-related health outcomes could be better explored [20,21,22,23,24].

The purpose of this study was to compare the predictive utility of some CCFIs (computed from varying formulation components) for child nutritional status (stunting, wasting, and underweight status) using a cross-sectional study data set obtained from a resource-constrained study setting in northern Ghana, a West African country in Sub-Saharan Africa (SSA).

## 2. Methods

### 2.1. Study Setting

The northern region of Ghana has the country’s highest prevalence rate of stunting (33% and 29%, respectively) according to the Ghana Demographic and Health Survey 2014 report [39] and the Multi Indicator Cluster Survey (MICS) 2017–2018 report [40]. The northern region also has the lowest score for the early childhood development index (ECDI) and early childhood education (ECE) for children between 36–49 months [40]. Children under five years constitute about 20% of the population in the northern region of Ghana. There are two main climatic seasons in the three erstwhile northernmost regions (northern region, upper west region, and upper east region), namely the short rainy season (May–August) and the long dry season (September–April). Subsistence agriculture remains the main source of livelihood for the majority of the northern region’s inhabitants, with a significant proportion involved in trading and relatively few highly educated and skilled professionals. October to December usually is the peak of the harvesting season, during which there is usually an abundance of indigenous staple foods, such as maize, sorghum, millet, and yams, except when occasional natural disasters derail the season’s harvest [41,42].

### 2.2. Study Design and Population of Interest

A community-based analytical cross-sectional design was used for this study conducted in June 2018. Together with their children (6–23 months), 634 mothers of reproductive age (15–49 years) were selected using a two-stage cluster sampling technique from stunting-endemic districts of the northern region of Ghana. There are several languages and dialects spoken in northern Ghana. However, the major languages spoken in the selected study districts are Dagbani (Dagombas), Gonja, and Nanumba. The study participants were drawn from 25 communities across five of the districts in the northern region with a relatively high prevalence of stunting (≥30%) amongst children under five years of age (Appendix A).

### 2.3. Sample Size and Sampling Procedures

The final sample size used for the statistical analyses of the study data was 581 and was determined from the standard formula for one-point sample estimation [43,44]. The primary outcome variable used to estimate the sample size was the population proportion of stunting prevalence as reported by the 2014 GDHS on the northern region (33.1%). An estimated 20% of the total population of the northern region are children under five years of age. With this 33.1% prevalence rate of chronic undernutrition (stunting), premised on 80% power and an absolute precision of 5% at the 95% confidence level, we estimated a sample size of *n* = 284. The required minimum sample size was estimated to be *n* = 568 based on the assumption of a correction factor of 2.0 due to the design effect for cluster sampling [29]. The calculated sample size was adjusted to 600 to account for a possible 5% non-response rate to cover limiting circumstances such as missing values, implausible values, damage, or loss of completed questionnaires, and withdrawal by some of the selected participants from the survey interviews and/or anthropometric measurements. A two-stage cluster sampling procedure was used to select the participants from each community in June 2018 as described in detail previously [11].

### 2.4. Data Collection Procedures and Instrument

A local field research team (data enumerators, supervisors, and data-entry clerks), recruited mostly from the northern region and provided with training prior to the pilot and definitive surveys, was led by the principal investigator to conduct an interviewer-administered survey of 634 mother–child pairs. The training exercise was to ensure high intra- and inter-rater reliability [45,46]. The training agenda included the purpose of the study and survey, sampling procedures, community, and household entry protocols, interviewing techniques, interpretation of questions from English language questionnaires into the local languages and vice versa, and assessment of anthropometric measurements (height/length and weight). A final revised version of the quantitative survey instrument was produced after combining and adapting extracts from the WHO interviewer-administered questionnaire used for assessing food intake in malnutrition studies, a food intake and access questionnaire, and an instrument adapted for a malnutrition study in northern Ghana [6,47,48]. The interviews were conducted in the local languages spoken by the study participants, with the interviewers using an English-language questionnaire. The interviewer-administered questionnaire was used to elicit self-reported dietary intake data (IYCF indicators) over a 24 h dietary recall (24HDR) period in addition to socio-demographic data and other relevant data, including the putatively proximal, intermediate, and distal determinants of undernutrition from the eligible mother–child pairs.

### 2.5. Construction of Various Composite Child Feeding Indices (CCFIs)

Self-reported dietary intake indicators of child breastfeeding and complementary feeding status and/or practices were used to construct five CCFIs with varying components and scoring guides as summarized in Table 2. The formula proposed by Ruel and Menon [14] was adapted together with other formulas applied in similar CCFI construction (Table 1). The potential variations in the inferences that could be drawn from the varying compositions of the formulation of CCFIs for the comparative analyses were premised on some theoretical and empirical assumptions [49]. CCFI 1 formulation was based on the seminal criterion (BF and CF) used by Ruel and Menon [14] in addition to its wide utilization in different study settings over the years. CCFI 2 formulation was based on the general hypothesis that child growth faltering occurs most significantly during complementary feeding (CF) [4]. Additionally, regarding exclusive breastfeeding (EBF) amongst infants in northern Ghana, though generally sub-optimal (≤90% threshold set by WHO), the prevalence rate appears not to be debilitating even though EBF is widely recognized as protective of child health and facilitative of optimal anthropometric growth [4,50]. CCFI 3 formulation was based on the premise that optimal child feeding practices (BF and CF) are broadly multidimensional (CCFI 1) and that CF is timing-sensitive (TICF) [4,7]. Additionally, diverse food intake (FVI) and the adequate intake of micronutrient-rich food sources of iron (Fe) and vitamin A (Vit A) especially from fruits and vegetables (F and V) are putatively facilitative for optimal child anthropometric growth [51]. CCFI 4 formulation was based on the assumption that besides the dimensions incorporated in the formulation of the classical CCFI by Ruel and Menon, adequate and high-quality protein intake mainly from animal food sources (AFS) and/or plant food sources (PFS) are most critical for optimal child anthropometric growth and should thus suffice for optimal child food and/or nutrient intake estimation [52,53,54]. CCFI 5 formulation was based on the assumption that besides the dimensions incorporated in the formulation of CCFI 1, all other food intake variables that constitute possible components or domains of optimal child feeding practices could be clinically and/or epidemiologically critical as determinants of optimal child anthropometric growth [55]. These components should thus be considered for incorporation in the child food and/or nutrient intake estimation using an appropriate dimension reduction statistical technique such as principal component analysis (PCA) [56,57]. During the statistical model specification, possible violation of the requisite underlying statistical assumptions especially multicollinearity and overfitting were also to be addressed [58,59,60].

### 2.6. Measures of Child Nutritional Status

Measures of child nutritional status were the outcome variables in this study. The heights/lengths and weights (anthropometric measurements) of the children and mothers were measured to determine nutritional status. The following measurements were used to calculate the anthropometric (bodily) indicators of underweight, stunting and wasting status, respectively: WAZ (weight-for-age), LAZ (length-for-age), and WLZ (weight-for-length: SD < −2) for the children and BMI (body mass index) for the mothers using the WHO Anthro Software Version 3. Computed Z-scores were based on the 2006 WHO growth standards, expressed as standard deviation units from the median value for the WHO growth reference groups [61]. Improbable Z-scores (scores falling outside the WHO flags): WLZ −5 to 5, LAZ −6 to 6, and WAZ −6 to 5 were excluded from the data set. The data on the anthropometric growth indicators (Z-scores) were exported into the IBM SPSS Statistics Software (Version 27) for further analyses. Children who fell below minus two standard deviations (−2 SD) from the median of the reference population for LAZ, WAZ, or WLZ were classified as stunted, underweight, or wasted, respectively. Mothers’ and children’s weights were measured using a standard electronic scale sensitive to the nearest 100 g (Seca 890). The recumbent length of each child was measured in a supine position to the nearest 0.1 cm with a portable Infantometer. This supine measurement was taken by placing each child on his or her back between the slanting sides, ensuring that the child’s head was placed gently against the fixed top end. The child’s knees were held down gently by the anthropometrist, while the movable foot-piece of the Infantometer was drawn up to touch the child’s feet at right angles to the legs. Some children who could stand appropriately were measured standing. The WHO Anthro software automatically converts height to length for children aged less than 24 months. For the mothers, height was measured in a standing position using a Seca microtoise stadiometer to the nearest 0.1 cm.

### 2.7. Independent Variables

The independent variables of interest in this study were the CCFIs in the continuous and categorical forms, constructed from various indicators of infant and young child feeding practices. The determinants (independent variables) of undernutrition in the northern region of Ghana in this study were classified into proximal or immediate, intermediate, or underlying and distal or basic factors based on an adapted version of the UNICEF hierarchical conceptual framework (Appendix A) as described in detail in a recent publication from the same research project [11].

### 2.8. Measurement of Infant and Young Child Feeding (IYCF) Practices

The child feeding practice indicators of this study were estimated from self-reported 24 h food recall (24HFR) [6,7]. Each mother selected for interview was asked to recall the number of times, in the last 24 h prior to the day of the survey, that her child had received any type of meal, snack, or drink (complementary feeding) from seventeen food groups and/or seven food groups as classified by the Food and Agriculture Organization of the United Nations (FAO) and WHO, respectively [7,48]. Timely introduction of complementary feeding (TICF) was defined as the commencement of complementary feeding (introduction of solid, semi-solid, and soft foods besides breast milk) at six months after birth. Minimum dietary diversity (MDD) was defined as the proportion of children (6–23 months) who were fed with meals made from food items or foods from at least four out of the seven food groups. Dietary diversity score (DDS) was determined as the score for the number of food groups out of the seven that each child had been fed from during the last 24 h, 7 being the highest and 0 being the lowest score. Minimum meal frequency (MMF) was defined as the proportion of children (6–23 months) who received the minimum recommended number of complementary feeds during the last 24 h prior to the survey. This measure (MMF) depends on the child’s age, as classified by the WHO (CF ≥ 2 times for 6–8 months and ≥3 times for 9–23 months plus snacks for breastfeeding children and ≥4 times in 24 h for non-breastfeeding children).

Minimum acceptable diet (MAD) was defined as the proportion of children who received both the MMF and MDD for their age category, as classified by the WHO. Intake of micronutrient-rich foods (MRF) was also estimated for vitamin A and iron (Fe), using the 17 food groups as classified by the FAO [48]. Children who received meals including items from at least one of the three iron-rich food groups were classified as having had adequate iron (Fe) intake. Children were classified as having received none (0), low (1–3), or high (more than four) vitamin A intake out of the seven vitamin-A-rich food groups in the self-reported 24 h food recall.

Appropriate complementary feeding (ACF) was a composite child feeding index (CCFI) constructed from TICF, MMF, and MDD as previously described [29,30]. ACF, as a composite index in this study, was defined as the proportion of children who received the MMF, MDD, and commenced complementary feeding at six months after birth (TICF) as recommended by WHO. MMF is conceived as a proxy measure of dietary energy intake adequacy from foods other than breastmilk, while MDD is putatively perceived to be a proxy measure of dietary quantity and quality. Age-appropriate timing for the introduction of complementary feeding is associated with positive nutritional outcomes for infants and young children under five [62].

### 2.9. Statistical Analyses

Responses from the interviewer-administered questionnaires (*n* = 634) were coded, entered, and screened using IBM SPSS Software (Version 27). Cases with missing values and/or implausible Z-scores were deleted from the dataset. The cleaned data set (*n* = 581) were evaluated for compliance with the key assumptions underlying multivariable regression analyses. These diagnostics included sample size adequacy, missing values, univariate outliers, multivariate outliers (Cook’s distance), normal distribution of residuals (P-P plot), linearity (scatter plots), homoscedasticity (equality or homogeneity of variance), multicollinearity (variance inflation factor (VIF)), and independence of observations or residuals (Durbin–Watson statistic) [63,64].

Descriptive analyses were performed to summarize the characteristics of the study participants, all the other covariates or factors (independent variables (IVs)), and prevalence of the dependent variables (DVs; nutritional status) using mean and standard deviation for the continuous variables, counts, and relative frequencies (percentages) for the categorical variables. Bivariate analyses were also performed to examine the distribution of each CCFI using ANOVA and *t*-test (normally distributed continuous form of DVs) and Pearson’s chi-square ((χ^2^) tests (categorical form of DVs). The means and standard deviations (SD) and relative frequencies were reported for the continuous and categorical forms of the CCFI component variables, respectively. Bivariate analyses were conducted also to determine associations (strength, direction, and significance) between each categorical independent variable (IV) and the categorical form of the dependent variables (less than −2 SD for stunting and wasting) using Pearson’s chi-square ((χ^2^) tests at *p* < 0.05. Pearson correlation coefficient (r) and simple linear regression were used to measure the strength, direction, and significance of the relationships between the covariates (continuous or non-categorical independent variables) and the continuous form of the DVs at *p* < 0.05 [65,66].

A predictive statistical modelling approach was used to analyze the association between each CCFI and the measures of child nutritional status (stunting, wasting, and underweight status), accounting for the effects of potential confounders (relevant factors and/or covariates) in the model specifications [67,68]. Multivariable binary logistic regression analyses and multiple linear regression analyses were conducted to determine the statistical significance of the associations between each CCFI and undernutrition (stunting, wasting, and underweight status) using the binary and continuous (Z-score) forms of the DVs, respectively. In order to obtain parsimonious statistical models, multicollinearity was assessed among the significant covariates and/or factors (IVs) selected from the bivariate analyses and literature search, using variance inflation factor (VIF) with a threshold of three (3) for the categorical variables and ten (10) for the non-categorical variables. The independent variables (IVs) found to be significantly associated with each of the dependent variables (DVs) in the bivariate analyses were used in the multivariable regression modelling at *p* < 0.05. However, explanatory variables that were considered to have clinical relevance from literature search and were also statistically significant at *p* < 0.10 were included in the multivariable regression models [65,69]. The multiple linear regression analyses were conducted using the general linear model (GLM) mode (univariate procedure) of SPSS because of its automated dummy-coding function of the categorical variables during the model specification. Multicollinearity was also assessed amongst the components used for the formulation of CCFI 5 by examining the correlation coefficients (r) with a threshold of 0.7 and VIF with a threshold of 3 for categorical variables and 10 for continuous variables. CCFI components that exhibited collinearity were excluded from the scoring criteria for CCFI 5 before the PCA.

The overall model performance (goodness of fit and calibration) were assessed using the Hosmer–Lemeshow goodness of fit test and the Nagelkerke R^2^ for the logistic regression models. The coefficient of determination (R^2^) was used as a measure of goodness of fit and parsimony for the multiple linear regression models [70,71]. Two-tailed statistical significance was reported at 95% confidence intervals (CI).

Content and construct validity of the five CCFIs were assessed qualitatively by the authors based on consensus, cognizant of inferences from the previous studies (Table 1). Internal consistency (reliability of homogeneity) of each of the CCFIs was examined using Cronbach’s alpha coefficient (α), mean inter-item correlation coefficient (MIICC), and/or Kuder–Richardson coefficient (K-R 20) where applicable [72,73,74].

Sensitivity (what-if) analyses of the study results were conducted to assess the robustness of the final parsimonious models based on the form of the CFFI used (continuous versus categorical scores in tertiles) and the exclusion of breastfeeding status from CCFI 1 formulation [75,76,77].

### 2.10. Ethical Clearance and Community Entry Protocols

In accordance with the requirements of the Helsinki Declaration, ethical clearance was obtained from the Ghana Health Service Ethics Review Committee (GHS-ERC: 011/11/17) in Accra, Ghana, and the Ethics Committee of Bielefeld University (EUB 2018-083) in Bielefeld, Germany. Community and household entry protocols for the study area were followed accordingly to gain access to the study participants. Written informed consent prior to enrolment was obtained from each mother by endorsement with a thumb print and/or signature on the consent forms provided after explaining the purpose and scope of the study to the participants in their native languages. Verbal consent was also obtained from the household heads in accordance with their cultural values and community norms

## 3. Results

Characteristics of study participants and prevalence of child nutritional status are as follows:

The majority of participating mothers (55.2%) were aged 25–34 years, farmers (55.6%), married (97.2%), and currently breastfeeding (96.4%). The majority of children (89.9%) weighed less than 2.5 kg at birth, with 32.2% being stunted (Table 3).

### 3.1. Distribution of the CFFI Scores

Mean group differences of CCFI scores between the child age groups were all statistically significant (*p* < 0.05). Of the five CCFIs, the child age group of 9–11 months had the highest median score (50th percentile) for CCFI 1, all the three age groups had the same median score for CCFI 2, child age group of 12–23 months had the highest median score for CCFI 3, child age group of 9–11 months had the highest median score for CCFI 4, and child age group of 12–23 months had the highest median score for CCFI 5 (Figure 1).

### 3.2. Comparative Analyses of the Predictive Utility of CCFIs for Child Nutritional Status

The final parsimonious predictive models of the CCFIs were arrived at with fairly similar potential confounders and/or covariates or factors adjusted for in the multivariable regression analyses following their selection from the bivariate analyses and/or literature search. These included the district of residence, religion, tribe (ethnicity), maternal age and height, child age group, usage of insect-treated nets (ITN), and the number of occupants per the child’s household, which were significantly associated with child stunting status in the bivariate analysis. Religion, tribe (ethnicity), marital status, maternal body mass index (BMI), child age group, child gender (sex), child health status (morbidity in the last two weeks and frequency of diarrhea in the last six months), child immunization status, frequency of prenatal care (PNC) services attended, and the source of power (energy) for household utility were significantly associated with child wasting status in the bivariate analysis. The district of residence, community, tribe, type of community (rural or urban), maternal BMI and height, child age group and gender, child health status (morbidity in the last two weeks), the source of power (energy) for household utility, and the number of people per room in the household were significantly associated with child underweight status in the bivariate analysis. Child gender, child’s recent morbidity status, and maternal BMI were consistently associated significantly with wasting status in the multivariable regression models, alongside with CCFI 1, 4, and 5.

None of the CCFIs in the continuous form was consistently predictive of all three measures of child nutritional status (stunting, wasting, and underweight) after controlling for potential confounders and/or correlates. The continuous forms of CCFI 1, 4, and 5 were significantly predictive of only wasting (WHZ) from the linear regression models after adjusting for potential confounders and/or correlates (Table 4).

The continuous form of the CCFIs were all not significantly predictive of child stunting (HAZ), wasting (WHZ), and underweight (WAZ) status in the bivariate regression models except CCFI 2 for stunting (logistic regression) and underweight status (linear regression); CCFI 3 for stunting (linear and logistic regression), wasting (linear regression), and underweight status (linear regression), CCFI 4 for wasting (linear regression) and underweight status (linear regression), and CCFI 5 for stunting (linear and logistic regression), wasting, and underweight status (linear regression). Only CCFI 3 and CCFI 5 were consistently associated with child nutritional status in the bivariate analyses irrespective of the type of regression method used except for the simple logistic regression analyses between CCFI 3 and wasting or underweight status and that between CCFI 5 and wasting status (Table 4).

There was consistency in the predictive outcomes of the continuous forms of the CCFIs irrespective of the choice of regression methods (linear or logistic) used except CCFI 1 for wasting (multivariable), CCFI 2 for stunting (bivariate) and underweight status (bivariate), CCFI 3 for wasting (bivariate) and underweight status (bivariate), CCFI 4 for wasting (multivariable and bivariate) and underweight status (bivariate), and CCFI 5 for wasting status (multivariable and bivariate) (Table 4).

There was consistency in the predictive outcomes of the continuous forms of the CCFIs for child nutritional status compared with the adjusted (multivariable) and unadjusted (bivariate) models except CCFI 1 for wasting (linear regression); CCFI 2 for stunting (logistic regression) and wasting status (linear regression); CCFI 3 for stunting (linear and logistic regression), wasting (linear regression), and underweight status (linear regression); CCFI 4 for underweight status (linear regression); and CCFI 5 for stunting (linear and logistic regression) and underweight status (linear and logistic regression). Only the continuous form of CCFI 1 exhibited Simpson’s paradox, that is, a spurious reversal of statistical significance, strength, and/or direction of association for the prediction of wasting (WHZ) in the linear regression model. The bivariate association between CCFI 1 and wasting was not statistically significant, but after controlling for the potential confounders and/or correlates, CCFI 1 became significantly associated with wasting (Table 4).

None of the CCFIs in the categorical form was consistently predictive of all three measures of child nutritional status (stunting, wasting, and underweight), irrespective of the regression method (linear or logistic), after controlling for potential confounders and/or correlates (Table 5).

The categorical form of each CCFI was not significantly predictive of child stunting (HAZ), wasting (WHZ), and underweight (WAZ) status in the bivariate regression models except CCFI 1 for stunting (linear and logistic regression) and underweight status (linear and logistic regression); CCFI 2 for underweight status (linear regression); CCFI 3 for stunting (linear and logistic regression), wasting (linear regression), and underweight status (linear regression); CCFI 4 for stunting (linear and logistic regression) and underweight status (logistic regression); and CCFI 5 for stunting (linear and logistic regression), wasting, and underweight status (linear regression). None of the categorical forms of the CCFIs was consistently associated with all the three measures of child nutritional status in the bivariate analyses for both linear and logistic regression model specifications (Table 5).

There was consistency in the predictive outcomes of the categorical forms of the CCFIs irrespective of the choice of regression methods (linear or logistic) used except CCFI 2 for underweight status (bivariate), CCFI 3 for wasting (bivariate) and underweight status (bivariate), CCFI 4 for underweight status (bivariate), and CCFI 5 for wasting (bivariate) and underweight status (bivariate) (Table 5).

There was consistency in the predictive outcomes of the categorical form of the CCFIs for child nutritional status compared with the adjusted (multivariable) and unadjusted (bivariate) models except CCFI 1 for stunting (linear and logistic regression) and underweight status (linear and logistic regression); CCFI 2 for underweight status (linear regression); CCFI 3 for stunting (linear and logistic regression), wasting (linear regression), and underweight status (linear regression); CCFI 4 for stunting status (linear and logistic regression); and CCFI 5 for stunting (linear and logistic regression), wasting (linear regression), and underweight status (linear regression). None of the categorical forms of the CCFIs exhibited Simpson’s paradox (Table 5).

### 3.3. Validity and Reliability Analyses of the Statistically Significant CCFIs

CCFI 1 and CCFI 5 had the lowest and highest Cronbach α values, respectively, in correspondence with the number and relevance of the test item components used in their construction (Table 6). Breastfeeding had a negative correlation with bottle feeding (BtF), Dietary Diversity Score (DDS), and meal frequency (MF) in the CCFI 1 formulation except for diverse food frequency intake recalled over a period (FFQ II, protein-rich foods and staples). Breastfeeding had a negative correlation with the test items BtF, DDS, MF, and iron-rich food sources (Fe) in the CCFI 4 formulation for except FFQ II. BF had a negative score (−0.472) from among the eight test items of the first component score of the PCA. Breastfeeding had a negative correlation with all the eight test item components used in the CCFI 5 formulation except for FFQ II. This first component of the PCA explained 31.90% of the variance in the construct (CCFI 5) developed from the 14 test items initially used for the factor analysis.

### 3.4. Sensitivity Analyses and Model Predictive Performance of Statistically Significant CCFIs

Only the CCFIs that were significant predictors of child nutritional status were assessed for the adjusted model performance and sensitivity to variations in child age data type options (continuous and categorical) used in the modeling. The adjusted coefficient of determination (adjR^2^) used to assess the trade-off between the goodness of fit and parsimony for the multiple linear regression models explained 6.7% of the variance in child wasting status for CCFI 1, 7.5% for CCFI 4, and 7.5% for CCFI 5 (Table 7).

## 4. Discussion

The purpose of this study was to compare the predictive utility of CCFIs (computed from varying formulation components) for child nutritional status (stunting, wasting, and underweight status). After controlling for potential confounders and/or correlates in the regression models, CCFI 5, which was constructed using a dimension reduction statistical technique, namely PCA, was found to have a relatively higher predictive utility for child wasting status compared to CCFI 1, which was constructed following the seminal formula of Ruel and Menon [14]. It was also significantly predictive of all the three measures of undernutrition (stunting, wasting, and underweight status) in the unadjusted models irrespective of the choice of regression method used, except for wasting (WHZ), in the bivariate logistic regression model specification. None of the CCFIs constructed was consistently predictive of all the three measures of child nutritional status (stunting, wasting, and underweight) after accounting for the potential confounders and/or correlates. A valid, reliable, standardized, and calibrated summary index has valuable and practical policy implications when the aggregated value is found to be epidemiologically interpretable and useful for public health decision making [78,79]. CCFI as a summary index could serve as a potent criterion for making rapid decisions, for instance, in emergency resource allocation and prioritizing public health nutrition interventions for the most chronically vulnerable populations. Such decision making would otherwise have to be carried out after evaluating many different predictive and mediating domains of optimal infant and young child food intake as a function of child undernutrition or nutritional status in developing countries.

In most of the studies that adopted the Ruel and Menon [14] approach to constructing the CCFIs and/or a modified version, the association between the CCFIs and the various anthropometric measures of child nutritional status (stunting, wasting, and underweight) were mostly statistically significant, especially for stunting [15,33,34], contrary to this study’s findings, except for wasting status (Table 1). However, some studies also reported non-significant multivariate associations between the CCFI and child nutritional status indicators similarly to the findings in this study [12,36,37,38]. In these studies, the formulation and construction of the CCFIs were similarly based on the five components used by Ruel and Menon. One possible reason why there seems to be instability or inconsistency in the association between the IYCF indicator components used for the CCFI construction and stunting (LAZ/HAZ) is that whereas IYCF indicators are one timepoint estimates of food intake practices, stunting is reflective of the effects of a long cumulative period of inadequate nutrition. Wasting is reflective of short-term effects of inadequate food intake. Ntab et al. [38] suggested that there was no significant association between the CCFI and stunting partly due to reverse causality between breastfeeding (BF) and stunting. BF was also negatively associated with the other components of the CCFIs constructed in this study, thus possibly accounting for the similarity in inference. The CCFI scores were significantly different for the three age categories in this study (Figure 1), as observed also by Ntab et al. [38]. However, Wondafrash et al. [12] was of the view that the narrow range of foods (low dietary diversity) similarly consumed by the children in this study could have affected the discriminatory power of the summary indices. Moursi et al. [36,37] posited that given that BF may be implicated in the reverse causality observed in its association with child nutritional status (stunting), CCFI construction and use for modelling should be disaggregated into BF- and CF-related variables.

To the best of our knowledge, no study has adopted the PCA approach to construct CCFI. However, similar epidemiological and public health studies that used dimension reduction as a statistical technique to generate a summary index for predictive modeling purposes found it to be robust and clinically useful [79,80,81].

Acute undernutrition also referred to as wasting status (WHZ/WLZ) is one of the key nutritional health indicators used in the construction of the Global Food Security Index (GFSI) and Global Hunger Index (GHI) for the classification of populations prone to food insecurity, hunger, and/or malnutrition. The consistency and suitability of the CCFIs especially CCFI 5 for predicting wasting status can also therefore be useful as a public health tool to rapidly identify undernourished children under five years under acute food shortage or emergency situations.

### 4.1. Validity and Reliability of Significantly Predictive CCFIs

The measurement validity (face, content, and criterion) of the CCFIs as constructed in this study were generally rated subjectively (qualitatively) as ranging between low and very good even though the test items of the sub-domains covered the relevant aspects of optimal child food intake, namely breastfeeding and complementary feeding [73,74]. CCFI 4 and 5 showed similarly good predictive and convergent validity (criterion) for child wasting status compared to the widely utilized and referenced summary index of optimal child food intake (CCFI 1) developed by Ruel and Menon [14]. These are suggestive of the suitability of the summary indices as being operationally representative or reflective of the concept of optimal child food intake for the purposes of measuring the construct even though the construct validity of the CCFIs were not quantitatively examined in this study.

The Cronbach alpha values of CCFI 1, 4, and 5 (α = 0.40, α = 0.60, and α = 0.80, respectively) were indicative of low to very good internal consistency as a measure of reliability (or homogeneity of the test items) of the construct representing optimal young child food intake [74,82,83]. The variations in the internal consistency, a measure of how well the test items or components of the summary indices steadily reflected the construct operationalized as a score of optimal child food intake, could be due to the number items per each CCFI formulated. The fewer the number of items, the lower the Cronbach α values [83]. Besides, the breastfeeding component of the CCFIs was negatively correlated with some of the other items as reported in similar studies [84].

The findings from the sensitivity analyses and model performance evaluations agree with studies that suggest that the choice of statistical method, form of the predictors in the regression models, and the selection criteria for inclusion or exclusion of one or more IVs in regression models could influence the inferences obtainable [56,85]. This therefore supports the call for standardized, transparent, and detailed reporting of methodological decisions in epidemiological studies as espoused by the STROBE advocates [86].

### 4.2. Strengths and Limitations

This study is the first attempt to compare the predictive utility of variously formulated composite child feeding indices (CCFIs) that reflect the multidimensionality of optimal child dietary intake for identifying infants and young children at risk of undernutrition (stunting, wasting, and underweight status). Even though the cross-sectional design used for this study precludes it from establishing causality, the reliability, validity, model performance, and sensitivity analyses conducted provides additional confidence in the study results. It also highlights some important conditions that potentially debunk or otherwise accentuate the robustness of the postulated association between CCFIs and child nutritional status measures.

Some possibly unmeasured confounders and residual confounding coupled with potential reporting bias (under- or overestimation from 24 HDR) and probable covariate selection bias may have suppressed or otherwise potentiated the effects of the CCFIs on child nutritional status [58,87]. The observed associations between the CCFIs and wasting status may not necessarily be generalizable to every resource-constrained study setting in LMICs but most likely are rather context-specific. The possibility of other forms of biases, such as endogeneity bias inherent in cross-sectional study designs, may not be entirely ruled out even though rigorous efforts were put into the study to address these possible drawbacks [88].

No internal and external model validation analyses were conducted; thus, the practical utility and generalizability of the CCFIs identified to be predictive of wasting status (WHZ) of children under five years cannot be proffered from this study. No clinical utility or practical epidemiological significance evaluation of the predictive models were conducted because none of the CCFIs was consistently predictive of all the three measures of child nutritional status (undernutrition) in this study after adjusting for potential confounders. No interaction analyses were conducted to assess the moderation effects of covariates such as child age and gender on the association between CCFI and wasting.

## 5. Conclusions and Recommendations

The predictive utility of CCFIs for child nutritional status is sensitive to the components and/or various measures of child feeding practices used in their formulation and construction. After adjusting for the potential confounders measured, CCFI 1, CCFI 4, and CCFI 5 were significant predictors of wasting status of children under five years.

The dimension reduction approach (PCA), which was used in formulating and constructing CCFI 5, is recommended for internal and external validation and possible adoption for wider applicability across heterogeneous study settings as the potentially optimum composite child feeding index usable for nutritional epidemiological studies among children under five years. Just as the hunger index (HI) and global food security index (GFSI) are used for ranking the vulnerability to malnutrition across communities, countries, and geographical regions, CCFI 5 could be used for a similar purpose.

More robust study designs (RCT, cohort, longitudinal, and case-control) and validation methods should be considered for exploration to address some of the limitations widely known to be inherent in cross-sectional design studies and self-reported dietary intake measurement instruments during the validation of these study findings. A comparative study of the effects of the choice of various regression methods (linear, logistic, quintiles, polynomial, and distributional approach) on the predictive utility of CCFIs for undernutrition, with or without the dichotomization (or categorization) of continuous data, is also recommended as explored by Sauzet et al. [75] in an observational epidemiological study.

## Figures and Tables

**Figure 1 ijerph-19-06621-f001:**
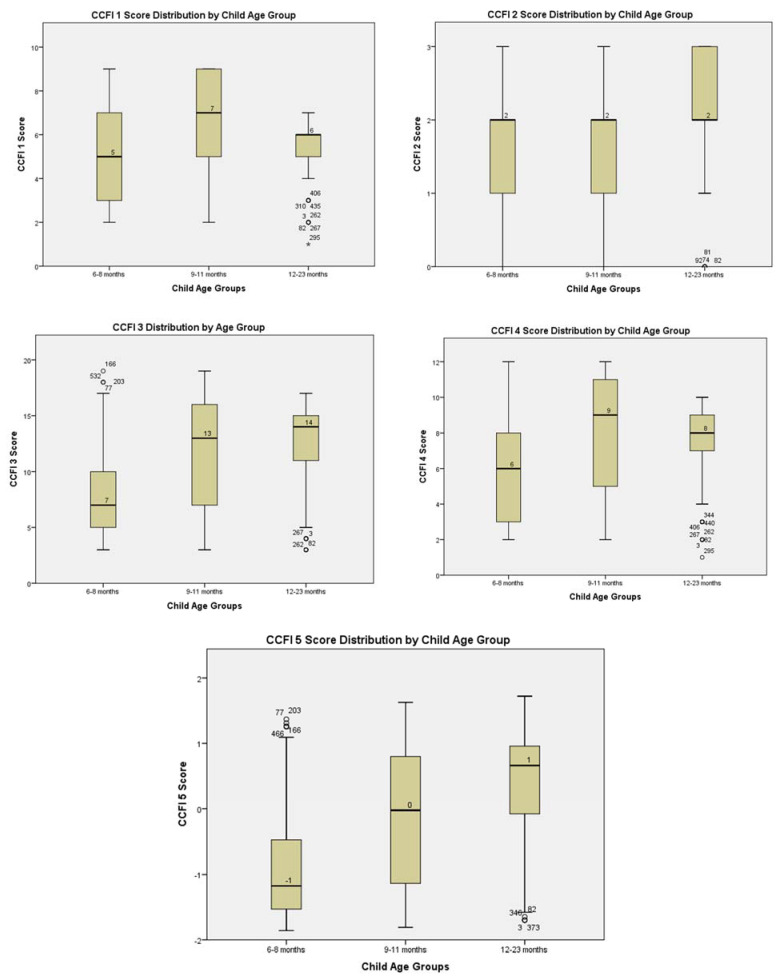
Box-whisker plots showing the distribution of CCFI scores by child age groups and univariate outliers (° regular outlier; * extreme outlier).

**Table 1 ijerph-19-06621-t001:** Summary of CCFI components used in the construction of some child feeding indices and the study characteristics.

Reference	CCFI Formulation Components	Age Range (Months)/Sample Size	CCFI Scoring Age Groups (Months)	Geography	Study Design/Statistical Analysis	Multivariable Study Findings	Remarks: Predictive Utility
Haq et al., 2020 [25]	1, 2, 4, 5	6–59/*n* = 800	6–8, 9–11, 12–36, 37–59	SEA/Rural/ Maldives	Cross-sectional/^φ^ Multiple Linear Regression	HAZ/LAZ *, WAZ *, WLZ **	Yes
Chaudhary et al., 2019 [26]	1 ^π^, 2 ^π^, 3, 4 ^π^	6–36/*n* = 210	6–9, 10–12, 13–36	SEA/Urban Slum/India	Cross-sectional/^φ^ Multiple Linear Regression	HAZ/LAZ *, WAZ *, WLZ *	Yes
Qu et al., 2017 [27]	1, 2, 3, 5, 9	6–35/ *n* = 12, 146	6–8.99, 9–11.99, 12–35.99	SEA/Rural/ China	Cross-sectional/^φφ^ Quantile Regression (Generalized Estimation Equation) and ^φφ^ Multiple Linear Regression	HAZ/LAZ *, WAZ *	Yes
Wondafrash et al., 2017 [12]	2, 3, 4, 5	6–18/*n* = 320	6–8, 9–12	SSA/Rural/ Ethiopia	Repeated Cross-sectional and Longitudinal/^φφ^ Multiple Linear Regression and ANCOVA	HAZ/LAZ **, WAZ **, WLZ **	No
Chowdhury, Rahman, and Khan, 2016 [28]	10a	6–23/*n* = 2373	6–11,12–17, 18–23	SEA/Urban and Rural/ Bangladesh	Cross-sectional/^φφ^ Multiple Binary Logistic Regression/ ^φφ^ Multivariable Multinomial Logistic Regression	Association with undernutrition not examined	Not examined
Saaka et al., 2016 [29]	10b	6–23/*n* = 778	6–11, 12–17, 18–23	SSA/Rural and Urban/Ghana	Cross-sectional/^φφ^ Multiple Binary Logistic Regression	Association with undernutrition not examined	Not examined
Kassa et al., 2016 [30]	10b	6–23/*n* = 611	6–11, 12–17, 18–23	SSA/Rural/ Ethiopia	Cross-sectional/^φφ^ Multiple Binary Logistic Regression	Association with undernutrition not examined	Not examined
Reinbott et al., 2015 [31]	1 ^π^, 2, 3, 4 ^π^, 5 ^π^	6–23/*n* = 803	6–8, 9–11, 12–23	SEA/Rural/ Cambodia	Cross-sectional/^φ^ Multiple Linear Regression and Non-linear Regression (Quadratic model)	HAZ/LAZ *	Yes, but weak
Lohia and Udipi, 2014 [15]	1 ^π^, 2, 3, 4, 5	6–24/*n* = 446	6–8.99, 9–11.99, 12–17.99, 18–24	SEA/Urban slum/India	Cross-sectional/^φ^ Multiple Linear Regression	HAZ/LAZ *, WAZ **, WLZ **, BAZ *, MUAC **	Yes
Ma et al., 2012 [16]	1, 4, 5, 11, 12	5–7/*n* = 180	6 (5–7), 12 (10–14), 18 (16–20)	SEA/Urban affluent city/ China	Longitudinal/^φφ^ Multiple Linear Regression and Stability Analysis	HAZ/LAZ *, WAZ *, WLZ **	Yes
Bork et al., 2012 [32]	1 ^π^, 3 ^π^, 4 ^π^, 14 ^π^	6–36/*n* = 1060	6–9, 9–12, 12–18, 24–36	SSA/Rural/ Senegal	Longitudinal/^φφ^ Multiple Linear Regression (Mixed Model)	HAZ/LAZ *	Yes
Khatoon et al., 2011 [33]	1, 2, 3, 4, 5	6–23/*n* = 259	6–8, 9–11, 12–23	SEA/Urban Hospital/ Bangladesh	Cross-sectional/^φφ^ Multiple Linear Regression	HAZ/LAZ *, WAZ **, WLZ **	Yes
Zhang et al., 2009 [34]	1, 2 ^π^, 3 ^π^, 4 ^π^, 5	6–11/*n* = 501	6–8, 9–11	SEA/Rural/ China	Cross-sectional/^φφ^ Multiple Linear Regression	HAZ/LAZ **, WAZ *, WLZ *	Yes
Garg et al., 2009 [35]	1, 2, 3 ^π^, 4 ^π^, 5, 9	6–12/*n* = 151	6–8, 9–12	SEA/Rural/ India	Cross-sectional/^φφ^ Multiple Linear Regression	HAZ/LAZ *, WAZ **, WLZ **	Yes
Moursi et al., 2009 [36]	1 ^π^, 2, 3 ^π^, 4 ^π^, 5	6–23/*n* = 1589	6–8, 9–11, 12–23	SSA/Urban/ Madagascar	Cross-sectional/^φφ^ Multiple Linear Regression	HAZ/LAZ **,WAZ **, WLZ **	No
Moursi et al., 2008 [37]	1 ^π^, 2, 3, 4, 5 ^π^	6–17/*n* = 363	6–8, 9–11, 12–17	SSA/Urban/ Madagascar	Longitudinal/^φφ^ Multiple Linear Regression	HAZ/LAZ **, WLZ **	No
Sawadogo et al., 2006 [17]	1 ^π^, 2 ^π^, 3 ^π^, 4 ^π^, 13 ^π^, 14 ^π^	6–35/*n* = 2466	6–11, 12–23, 24–35	SSA/Rural/ Burkina Faso	Cross-sectional/^φφ^ Multiple Linear Regression	HAZ/LAZ *, WLZ *	Yes
Ntab et al., 2005 [38]	1, 2, 3, 4, 15, 16, 17	12–42/*n* = 500	12–42	SSA/Rural/ Senegal	Cross-sectional/^φφ^ Multiple Linear Regression	HAZ/LAZ **	No
Ruel and Menon, 2002 [14]	1, 2, 3, 4, 5	6–36/*n* = (257, 341, 234, 459, 203, 303, 709)	6–9, 9–12, 12–36	LA/Rural and Urban/Bolivia, Colombia, Guatemala, Nicaragua, Peru	Cross-sectional/^φφ^ Multiple Linear Regression	HAZ/LAZ *	Yes, first study to construct such an index

1. Breastfeeding Status (currently, frequency); 2. Bottle Feeding; 3. Dietary Diversity I (Twenty-Four Hour Dietary Recall, 24HDR of consumption from 7 food groups); 4. Age-Appropriate Complementary Meal Frequency (per day)-(CF ≥ 2 times for 6–8 months and ≥3 times for 9–23 months plus snacks for breastfeeding children and ≥4 times in 24 h for non-breastfeeding children); 5. Food Frequency Questionnaire, FFQ I (7-day recall of consumption of a variety of food groups); 6. FFQ II (1-day recall) as substitute for FFQ I using only protein intake source- animal source protein (ASP) (yes = 2, no = 0) or plant source protein (PSP) (yes = 1, no = 0); 7. Fe, iron-rich food source—animal food source (yes = 2, no = 0), iron-fortified foods (yes = 2, no = 0), plant food source (yes = 1, no = 0); 8. Fruits and Vegetables—vitamin A-rich (yes = 2, no = 0), others (yes = 1, no = 0); 9. Timely Introduction to Complementary Feeding (TICF), <4 months or ≥9 months = 0, 4–5 months = 1, and 6–8 months = 2; 10a. Dimension Index from 20 food intake questions administered to mothers during Demographic and Health Survey (DHS); 10b. Appropriate Complementary Feeding (ACF), from meeting three complementary feeding-related criteria (TICF, MDD and MMF) score (yes = 1, no = 0); 11. Food Consistency, ≤6 months (gruel-like food = 1, semi-solid = 2), 12–18 months (gruel-like food = 0, semi-solid food = 1 and solid food = 2); 12. Dietary Diversity II (7-day recall of consumption from 8 food groups); 13. Specific unhealthy/healthy foods and drinks (snacks, sweetened and carbonated beverages); 14. Food Variety Index, FVI (1-day recall of consumption of a variety of food groups); 15. FFQ III (7-day recall) Protein intake-Meat; 16. FFQ IV (7-day recall) Protein intake—milk; 17 FFQ V (7-day recall) Protein intake—Fish; ^π^ Individual Components of CCFI Significant in Bivariate Analysis with LAZ/HAZ; SSA, Sub-Saharan Africa; SEA, South East Asia; LA, Latin America; Multivariable Study Findings—* Significant association and ** Non-significant association; ^φ^ CCFI treated as continuous variable; ^φφ^ CCFI treated as categorical variable; ANCOVA, Analysis of Covariance.

**Table 2 ijerph-19-06621-t002:** Formulas (components and criteria) applied for CCFI construction and scoring.

CCFIs	Components	Age Group Scoring			Remarks
		6–8	9–11	12–23	
CCFI 1	1, 2, 3, 4, 5	^1^ (Yes = 2, No = 0), ^2^ (Yes = 0, No = 1), ^3^ (0 fdg = 0, 1–3 fdgs = 1, ≥4 fdgs = 2), ^4^ (0 meal/day = 0, 1 meal/day = 1, 2^+^/day = 2), ^5^ (ASP, Yes = 2, No = 0; PSP, Yes = 1, No = 0)	^1^ (Yes = 2, No = 0), ^2^ (Yes = 0, No = 1), ^3^ (0 fdg = 0, 1–3 fdgs = 1, ≥4 fdgs = 2), ^4^ (0 meal/day = 0, 1–2 meals/day = 1, 3^+^/day = 2), ^5^ (ASP, Yes = 2, No = 0; PSP, Yes = 1, No = 0)	^1^ (Yes = 1, No = 0), ^2^ (Yes = 0, No = 1), ^3^ (0 fdg = 0, 1–3 fdgs = 1, ≥4 fdgs = 2),^4^ (0–1 meal/day = 0, 2–3 meals/day = 1, 4^+^/day = 2), ^5^ (ASP, Yes = 1, No = 0; PSP, Yes = 3, No = 0)	FFQ I (7-day recall: diverse food intake) was substituted with ^5^ FFQ II instead, unlike the classical formula used by Ruel and Menon.
Maximum total score		10 points	10 points	10 points	
CCFI 2	9	ACF (yes = 1, no = 0) if TICF, MDD, and MMF are all yes	ACF (yes = 1, no = 0) if TICF, MDD, and MMF are all yes	ACF (yes = 1, no = 0) if TICF, MDD, and MMF are all yes	Only CF-related core IYCF indices were used.
Maximum total score		1 point	1 point	1 point	
CCFI 3	1, 2, 3, 4, 5, 6, 7, 8, 10	Same as CCFI 1 plus ^6^ Fe (AFS (yes = 2, no = 0), ^α^ IFF, PFS (yes = 1, no = 0)), ^7^ Fruits and Vegetables (VitA-rich (yes = 2, no = 0), Other F and V (yes = 1, no = 0)), ^8^ TICF (0 or 1), ^10^ FVI (0, 1 or 2)	Same as CCFI 1 plus ^6^ Fe (AFS (yes = 2, no = 0), ^α^ IFF, PFS (yes = 1, no = 0)), ^7^ Fruits and Vegetables (VitA-rich (yes = 2, no = 0), Other F and V (yes = 1, no = 0)), ^8^ TICF (0 or 1), ^10^ FVI (0, 1 or 2)	Same as CCFI 1 plus ^6^ Fe (AFS (yes = 2, no = 0), ^α^ IFF, PFS (yes = 1, no = 0)), ^7^ Fruits and Vegetables (VitA-rich (yes = 2, no = 0), Other F and V (yes = 1, no = 0)), ^8^ TICF (0 or 1), ^10^ FVI (0, 1 or 2)	CCFI 1 plus intake of micronutrient-rich foods (MRF), TICF, and intake of varieties of foods (FVI).
Maximum total score		20 points	20 points	20 points	
CCFI 4	1, 2, 3, 4, 5, 6	Same as CCFI 1 plus ^6^ Fe (AFS (yes = 2, no = 0), PFS (yes = 1, no = 0))	Same as CCFI 1 plus ^6^ Fe (AFS (yes = 2, no = 0), PFS (yes = 1, no = 0))	Same as CCFI 1 plus ^6^ Fe (AFS (yes = 2, no = 0), PFS (yes = 1, no = 0))	CCFI 1 plus predominant source of iron intake (animal or plant)
Maximum total score		13 points	13 points	13 points	
CCFI 5	All possible CCFI components	CCFI 3 plus all the other possible components not exhibiting multicollinearity.	CCFI 3 plus all the other possible components not exhibiting multicollinearity.	CCFI 3 plus all the other possible components not exhibiting multicollinearity.	Excluded collinear components. 1st principal component used.
Maximum total score		Eigenvalue (Appendix B)	Eigenvalue (Appendix B)	Eigenvalue (Appendix B)	

^1^ Breastfeeding Status (currently; yes = 1, no = 0); ^2^ Bottle Feeding (yes = 1, no = 0); ^3^ Dietary Diversity Score (DDS)—Twenty-Four Hour Dietary Recall, 24HDR of consumption from 7 food groups, fdgs); ^4^ Age-Appropriate Complementary Meal Frequency (MMF) per day—(CF ≥ 2 times for 6–8 months and ≥3 times for 9–23 months plus snacks for breastfeeding children and ≥4 times in 24 h for non-breastfeeding children); ^5^ FFQ II (1-day recall) as substitute for food frequency questionnaire, FFQ I (7-day recall of diverse food intake; protein-rich foods and staples) using only protein intake source for scoring—animal source protein (ASP) or plant source protein (PSP); ^6^ Fe, iron-rich food source—animal food source, AFS (yes = 2, no = 0), iron-fortified foods, ^α^ IFF (excluded), plant food source, PFS (yes = 1, no = 0); ^7^ Fruits and Vegetables (F and V)—vitamin A-rich (yes = 2, no = 0), other F and V (yes = 1, no = 0); ^8^ Timely Introduction to Complementary Feeding (TICF), <4 months or ≥9 months = 0, 4–5 months = 1, and 6–8 months = 2; 9. Appropriate Complementary Feeding (ACF), from meeting three complementary feeding (CF)-related criteria (TICF, MDD, and MMF) score (yes = 1, no = 0); ^10^ Food Variety Index, FVI (1-day/24 h recall of consumption of a variety of 17 individual foods or food groups; 0–17 and converted into age-based terciles; Low FVI = 0, Medium FVI = 1, and High FVI = 2 guided by Bork et al., 2012 [32]).

**Table 3 ijerph-19-06621-t003:** Maternal and child characteristics (*n* = 581).

Characteristics	Frequency (*n*)	%
**Maternal age ****		
15–24 years	136	23.4
25–34 years	321	55.2
35–49 years	124	21.3
**Marital status**		
Unmarried	16	2.8
Married	565	97.2
**Maternal height**		
160 cm and above	282	48.5
Below 160 cm	299	51.5
**Occupation**		
Trader/vendor/manual laborer	166	28.6
Farmer	323	55.6
Vocational/skilled service worker	48	8.3
Unemployed	44	7.6
**Currently breastfeeding**		
Yes	560	96.4
No	21	3.6
**Child age ***		
6–11 months	242	41.7
12–17 months	185	31.8
18–23 months	154	26.5
**Child gender**		
Male	301	51.8
Female	280	48.2
**Child** **’s nutritional status**		
Stunting	193	33.2
Wasting	82	14.1
Underweight	157	27.0
**Child’s birth weight (*n* = 274)**		
Less than 2.5 kg	246	89.8
More than 2.5 kg	28	10.2

* Mean child age was 13.25 ± 5.09 months. ** Mean maternal age (±standard deviation (SD)) was 29.31 ± 6.40 years.

**Table 4 ijerph-19-06621-t004:** Predictive utility of continuous forms of CCFIs for child nutritional status.

CCFIs (Continuous)	Child Nutritional Status		
	Stunting	Wasting	Underweight
CCFI 1	* HAZ_α_, * HAZ_β_, * HAZ_π_, * HAZ_Σ_	** WHZ_α_, * WHZ_β,_ * WHZ_π_, * WHZ_Σ_	* WAZ_α_, * WAZ_β,_ * WAZ_π_, * WAZ_Σ_
CCFI 2	* HAZ_α_, * HAZ_β_, * HAZ_π_, ** HAZ_Σ_	* WHZ_α_, * WHZ_β,_ * WHZ_π_, * WHZ_Σ_	* WAZ_α_, * WAZ_β,_ ** WAZ_π_, * WAZ_Σ_
CCFI 3	* HAZ_α_, * HAZ_β_, ** HAZ_π_, ** HAZ_Σ_	* WHZ_α_, * WHZ_β,_ ** WHZ_π_, * WHZ_Σ_	* WAZ_α_, * WAZ_β,_ ** WAZ_π_, * WAZ_Σ_
CCFI 4	* HAZ_α_, * HAZ_β_, * HAZ_π_, * HAZ_Σ_	** WHZ_α_, * WHZ_β,_ ** WHZ_π_, * WHZ_Σ_	* WAZ_α_, * WAZ_β,_ ** WAZ_π_, * WAZ_Σ_
CCFI 5	* HAZ_α_, * HAZ_β_, ** HAZ_π_, ** HAZ_Σ_	** WHZ_α_, * WHZ_β,_ ** WHZ_π_, * WHZ_Σ_	* WAZ_α_, * WAZ_β,_ ** WAZ_π_, ** WAZ_Σ_

α, Multiple Linear Regression; β, Multiple Binary Logistic Regression; π, Simple Linear Regression; Σ, Simple Logistic Regression; HAZ (Stunting); WHZ (Wasting); WAZ (Underweight); ** Statistically significant at *p* < 0.05; * Not statistically significant at *p* < 0.05; LinReg, Linear Regression; LogReg, Logistic Regression.

**Table 5 ijerph-19-06621-t005:** Predictive utility of categorical forms of CCFIs for child nutritional status.

CCFIs (Categorical)	Child Nutritional Status		
	Stunting	Wasting	Underweight
CCFI 1	* HAZ_α_, * HAZ_β_, ** HAZ_π_, ** HAZ_Σ_	* WHZ_α_, * WHZ_β,_ * WHZ_π_, * WHZ_Σ_	* WAZ_α_, * WAZ_β,_ ** WAZ_π_, ** WAZ_Σ_
CCFI 2	* HAZ_α_, * HAZ_β_, * HAZ_π_, * HAZ_Σ_	* WHZ_α_, * WHZ_β,_ * WHZ_π_, * WHZ_Σ_	* WAZ_α_, * WAZ_β,_ ** WAZ_π_, * WAZ_Σ_
CCFI 3	* HAZ_α_, * HAZ_β_, ** HAZ_π_, ** HAZ_Σ_	* WHZ_α_, * WHZ_β,_ ** WHZ_π_, * WHZ_Σ_	* WAZ_α_, * WAZ_β,_ ** WAZ_π_, * WAZ_Σ_
CCFI 4	* HAZ_α_, * HAZ_β_, ** HAZ_π_, ** HAZ_Σ_	* WHZ_α_, * WHZ_β,_ * WHZ_π_, * WHZ_Σ_	* WAZ_α_, * WAZ_β,_ * WAZ_π_, ** WAZ_Σ_
CCFI 5	* HAZ_α_, * HAZ_β_, ** HAZ_π_, ** HAZ_Σ_	* WHZ_α_, * WHZ_β,_ ** WHZ_π_, * WHZ_Σ_	* WAZ_α_, * WAZ_β,_ ** WAZ_π_, * WAZ_Σ_

α, Multiple Linear Regression; β, Multiple Binary Logistic Regression; π, Simple Linear Regression; Σ, Simple Logistic Regression; HAZ (Stunting); WHZ (Wasting); WAZ (Underweight); ** Statistically significant at *p* < 0.05; * Not statistically significant at *p* < 0.05; LinReg, Linear Regression; LogReg, Logistic Regression.

**Table 6 ijerph-19-06621-t006:** Reliability and validity assessments of CCFIs.

CCFIs	Reliability		Validity		
	Cronbach’s α	α If Item ^#^ Deleted	Face	Content	Criterion ^@^ (Wasting)
CCFI 1	0.40	0.56	Good	Medium	Fairly good
CCFI 4	0.60	0.71	Very good	High	Good
CCFI 5	0.80	0.86	Excellent	Very high	Very good

^#^, Cronbach’s α if breastfeeding (BF) test item component was deleted from the construction of CCFI; ^@^, Predictive criterion validity with CCFI 1 as reference formula for construction of the summary index similarly developed by Ruel and Menon (2002) [14].

**Table 7 ijerph-19-06621-t007:** Predictive model performance indices and statistics.

Significant CCFIs	Effect Size		*F*-Statistic	*p*-Value	95% CI
	R^2^	adjR^2^			
CCFI 1	0.098	0.067	3.994	0.046	−0.126, −0.001
CCFI 4	0.102	0.075	6.996	0.008	−0.095, −0.014
CCFI 5	0.102	0.075	7.007	0.008	−0.265, −0.039

R^2^, coefficient of determination; AdjR^2^, adjusted coefficient of determination; F-statistic value; *p*-value, statistical significance at *p* < 0.05; CI, confidence interval.

## Data Availability

Data are available from corresponding author upon reasonable request.

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
