# Peer review of "Predictive Utility of Composite Child Feeding Indices (CCFIs) for Child Nutritional Status: Comparative Analyses for the Most Suitable Formula for Constructing an Optimum CCFI"

_ijerph, 2022, doi:10.3390/ijerph19116621_

Round 1

Reviewer 1 Report

I have just had a chance to review author responses and they were not incorporated in the manuscript, but have been adequately addressed in the write-up given that most of my queries and suggestions were methodological in nature. The authors have provided adequate explanations with relevant citations for why they have not made major methodological changes and I am satisfied with their response and revisions are adequate, overall.

Author Response

Thank you very much for your positive feedback.

Reviewer 2 Report

Thank you very much for allowing me to review the article entitle “Predictive utility of composite child feeding indices (CCFIs) for child nutritional status: a comparative cross-sectional study of CCFIs constructed using different formulas.” (ijerph-1708593).

The aim of this study was to identify the most suitable CCFIs (computed from varying formulation components) for child nutritional status among them for possible standardization, validation, and adoption by nutritional health researchers.Basically “The purpose of this study was to compare the predictive utility of some CCFIs (computed from varying formulation components) for child nutritional status (stunting, wasting, and underweight status) using a cross-sectional study data set obtained from a resource-constrained study setting in northern Ghana, a West African country in Sub-Saharan Africa
(SSA).”

They used a cross-sectional survey study on a cluster sampling (581 mothers-child pairs). They found that the dimension reduction approach which was used in constructing CCFI 5 could be recommended for standardization, validation, and possible adoption for wider applicability across heterogeneous population settings, as an optimum CCFI usable for nutritional epidemiological studies among children under five years.

Comments:

The manuscript has corrections, so, although I am reviewing it for the first time, I can verify that it has been revised and that the authors have incorporated recommendations.

The title of the work is not informative of the objective of the study; I recommend that the title of the work adjusts to the objective.

The introduction raises the interest of the topic but reports on the peculiarities of children in West African country in Sub-Saharan Africa (SSA). It should provide information on the situation of child nutrition in this area.

Table 1 of summary of CCFI components used in the construction of some child feeding indices, I suggest that you go to results since it is part of the contributions of the work, the review of these indices.

Material and methods: Include the revision of the indexes carried out in the bibliography, indicating how it has been carried out.

The information on the prevalence rate of stunting is from 2014 and we are in 2022, this information should be updated. The same goes for the rest of the literature.

The study was carried out in June 2018. The calculation of the sample size must be revised if the most updated prevalence of stunting changes.

What type of sampling was carried out? Were age and sex criteria used so that they were representative of the population? Is the sample homogeneous?

The results are well presented.

Discussion: It is well presented, but among the limitations is the use of a cross-sectional design and not a longitudinal design, which would be more appropriate.

It should be noted that the results refer to the population in the West African country in Sub-Saharan Africa (SSA) studied, since no other type of population has been prayed for.

Author Response

Thank you very much for allowing me to review the article entitle “Predictive utility of composite child feeding indices (CCFIs) for child nutritional status: a comparative cross-sectional study of CCFIs constructed using different formulas.” (ijerph-1708593).

The aim of this study was to identify the most suitable CCFIs (computed from varying formulation components) for child nutritional status among them for possible standardization, validation, and adoption by nutritional health researchers. Basically “The purpose of this study was to compare the predictive utility of some CCFIs (computed from varying formulation components) for child nutritional status (stunting, wasting, and underweight status) using a cross-sectional study data set obtained from a resource-constrained study setting in northern Ghana, a West African country in Sub-Saharan Africa (SSA).”

They used a cross-sectional survey study on a cluster sampling (581 mothers-child pairs). They found that the dimension reduction approach which was used in constructing CCFI 5 could be recommended for standardization, validation, and possible adoption for wider applicability across heterogeneous population settings, as an optimum CCFI usable for nutritional epidemiological studies among children under five years.

Comments:

The manuscript has corrections, so, although I am reviewing it for the first time, I can verify that it has been revised and that the authors have incorporated recommendations.

The title of the work is not informative of the objective of the study; I recommend that the title of the work adjusts to the objective.

Response: Thank you for your recommendation. The title has been revised to read “Predictive utility of composite child feeding indices (CCFIs) for child nutritional status: comparative analyses for the most suitable formula for constructing an optimum CCFI’’

The introduction raises the interest of the topic but reports on the peculiarities of children in West African country in Sub-Saharan Africa (SSA). It should provide information on the situation of child nutrition in this area.

Response: Thank you for the suggestion. The focus of the introduction is on the CCFI construction from IYCF measures and its potential universal application for predicting child nutritional status. Information on child nutrition peculiar to the study area (Northern Region of Ghana) was provided under the ‘Methods’ section as recommended for item 5 of STROBE-Nut (Lachat C et al. (2016) STrengthening the Reporting of OBservational studies in Epidemiology – Nutritional Epidemiology (STROBE-nut): an extension of the STROBE statement. Plos Medicine 13(6) http://dx.doi.org/10.1371/journal.pmed.1002036  

Table 1 of summary of CCFI components used in the construction of some child feeding indices, I suggest that you go to results since it is part of the contributions of the work, the review of these indices.

Response: Thank you for your suggestion. Table 1 is a summarized tabular literature review of variously formulated CCFIs and their use for predicting child nutritional status. It adds to the background information and rationale for comparing CCFIs constructed differently in previous studies and thus inspired our actual study. Since our objective was not to search for literature for a comprehensive, scoping, or systematic review, the summary information synthesized from literature on CCFIs would not be suitable as part of our original/focal study results. Thank you.

The information on the prevalence rate of stunting is from 2014 and we are in 2022, this information should be updated. The same goes for the rest of the literature.

Response: Thank you for your observation. At the time of data collection in June 2018, the most up to date estimate of stunting was from the Ghana Demographic and Health Survey (GDHS 2014) which was used for the sample size calculations.

The study was carried out in June 2018. The calculation of the sample size must be revised if the most updated prevalence of stunting changes.

Response: Thank you. An adjustment of the sample size calculation in line with an updated prevalence of stunting for data already collected in June 2018 does not come across as best practice. Would that then not mean an adjustment of the sample size to match the current population of children under five years in the study area in 2022 which was part of the basis for calculating the minimum sample size of 568?

In any case sample size estimation based on population proportion of the outcome variable of interest if unknown can assume a probability of 50% prevalence. Consequently, if there is no information available to approximate p, then p=0.5 can be used to generate the most conservative, or largest, sample size.

Using the formula for determining sample size:

where Z is the value from the standard normal distribution reflecting the confidence level that will be used (e.g., Z = 1.96 for 95%) and E is the desired margin of error. p is the proportion of expected or known prevalence of the outcome variable in the population. When planning a study to generate a 95% confidence interval for the unknown population proportion, p, 50% can be assumed.

Reference: Suresh, K., & Chandrashekara, S. (2012). Sample size estimation and power analysis for clinical research studies. Journal of human reproductive sciences5(1), 7–13. https://doi.org/10.4103/0974-1208.97779 (Retraction published J Hum Reprod Sci. 2015 Jul-Sep;8(3):186)

What type of sampling was carried out? Were age and sex criteria used so that they were representative of the population? Is the sample homogeneous?

Response: A two-stage random sampling procedure was used in the study, stage 1 was cluster random sampling and stage 2 was systematic random sampling. Detailed description of the sampling procedure is referenced in a previous related study conducted by the authors.

Anin, S.K.; Saaka, M.; Fischer, F.; Kraemer, A. Association between Infant and Young Child Feeding (IYCF) Indicators and the Nutritional Status of Children (6-23 Months) in Northern Ghana. Nutrients 2020, 12, doi:10.3390/nu12092565.

Yes children under five years were the target population. The sample was randomly collected giving equal and independent chance of selection to each child of any gender.

The results are well presented.

Response: Thank you

Discussion: It is well presented, but among the limitations is the use of a cross-sectional design and not a longitudinal design, which would be more appropriate.

Response: Thank you for the observation. We did recommend that more robust study designs and methodological procedures should be considered in future validation studies of the study findings.

It should be noted that the results refer to the population in the West African country in Sub-Saharan Africa (SSA) studied, since no other type of population has been prayed for.

Response: Thank for the observation. That was noted thus we indicated that the results may not necessarily be generalizable to every resource-constrained study setting in LMICs but most likely are rather context-specific. This can only be verified from future validation studies in other study settings.

This manuscript is a resubmission of an earlier submission. The following is a list of the peer review reports and author responses from that submission.

Round 1

Reviewer 1 Report

Thank you for the opportunity to review this manuscript. 

Specific comments for this manuscript begin with: 

  • Section 2.8: 
    • Can you clarify if this was a multiple pass or single pass 24 hour recall? Based on reading, it appears that the recall was conducted only on 1 day. Can you clarify if this instrument was validated to the local context? If so, please indicate the validation process.
    • MDD needs to be cited https://inddex.nutrition.tufts.edu/data4diets/indicator/minimum-dietary-diversity-mdd
    • How does one account for introduction of liquids other than BF?
  • Section 2.9  
    • Were directed acyclic graphs and/or any other conceptual frameworks used to define covariates in modelling in addition to significance in bivariate analyses and examination of the literature? How was confounding accounted for? 
    • Why was an EFA/CFA analysis not conducted to assess content and construct validity for scores? 

      https://support.sas.com/resources/papers/proceedings/proceedings/sugi31/200-31.pdf

  • Discussion:
    • I would agree with this point and would suggest that breastfeeding be examined separately in addition to CF. Given that the age range for this study is 6 mo and >, it is important to consider when CFs are typically introduced in this study population, given that recommendations may not be consistently followed and may be greatly impacted by month of birth and season. 
    • Same comment as above, consider using EFA/CFA for dimension reduction. Scale construction and variable construction is typically done using these methods. 
    • It would be helpful for authors to provide a clear elucidation of other scores and measures to assess child complementary feeding practices, including the FANTA project indicators and how they apply to your study. There seems to be some level of recursive repetition and I remain unclear on how this particular CCFI5 score will be particularly useful in emergency contexts given that Ghana is not a humanitarian context. 
    • Overall methods for construct and content validity are appropriate based on the literature, my only suggestion is to explain and/or examine the use of factor analysis for dimension reduction in addition to the other linear and logistic methods used for score/index construction. 

Reviewer 2 Report

Thank you very much to lowing me to review the article entitle “Predictive utility of composite child feeding indices (CCFIs) for child nutritional status: a comparative cross-sectional study of CCFIs constructed using different formulas.” (ijerph-1583867).

The aim of this study was to compare the predictive utility of some CCFIs (computed from varying formulation components) for child nutritional status (stunting, wasting, and underweight status) using a cross-sectional study data set obtained from a resource-constrained study setting. in northern Ghana, a West African country in Sub-Saharan Africa (SSA).

Comments:

The objective of the study should be stated clearly and specifically in the abstract, as well as the methodology applied and the sample size used.

The introduction is appropriate. Table 1 should present the studies in chronological order or by country where the study was carried out.

In the methodology section, in point 2.1 Study setting, the 3 studied areas should be presented (Northern Ghana, a West African country in Sub-Saharan Africa), and the time in which the study is carried out in each of them. The information that currently appears I recommend that you go to the introduction.

In the methodology section 2.2, a figure should be added that informs us of the selection of the sample of the inclusion and exclusion criteria and if it´s a proportional probabilistic sample or a convenience sample, this section is essential in relation to the results. Therefore, it should be clarified exactly how the sample has been collected and how it is constituted, given that there are 3 participating areas.

The calculation of the sample size assumes a uniform prevalence in the 3 locations, is this correct?

The sample of 634 mother and child from which area do they come from?

Was weight and height asked or measured? This is very important since there is scientific evidence showing that when asked the error is quite high (2.6). Reference 11 cannot be used to explain how it has been done, it must be explained in this work.

In the material and methods section 2.5, there is a lot of information that should be in the introduction and in the discussion, it does not correspond to material and methods.

As it appears in section 2.9, it is not 634 pairs of mother and child that are studied, but 581.

The results should be presented according to the 3 areas studied and focused on the objective.

it is not well understood how to get to table 6 and 7. Tables four and 5 should contain data

General comments: This is a very interesting study but with great weaknesses since the sections in the introduction are not respected it must be reported on material and methods it must be said what has been used as a result the results of the study according to the hypothesis and objective.

However, in this work there is a mixture in all the sections that should be corrected.

The most important weakness is that an objective is set between zones but only one is worked on, several couples (634) mother and son are said and then we find another (581), there is talk of 3 zones but only one is worked on with a.

In addition, there is talk of another work that we must review to understand this, therefore it seems like what has remained of the other work (reference 11).